# Iron status influences mitochondrial disease progression in Complex I-deficient mice

CJ Kelly, Reid K Couch, Vivian T Ha, Camille M Bodart, Judy Wu, Sydney Huff, Nicole T Herrel, Hyunsung D Kim, Azaad O Zimmermann, Jessica Shattuck, Yu-Chen Pan, Matt Kaeberlein*, Anthony S Grillo*†

Department of Laboratory Medicine & Pathology, University of Washington, Seattle, United States

**\*For correspondence:**
kaeber@uw.edu (MK);
grilloas@uc.edu (ASG)

**Present address:** †Department of Chemistry, University of Cincinnati, Cincinnati, United States

**Abstract** Mitochondrial dysfunction caused by aberrant Complex I assembly and reduced activity of the electron transport chain is pathogenic in many genetic and age-related diseases. Mice missing the Complex I subunit NADH dehydrogenase [ubiquinone] iron-sulfur protein 4 (NDUFS4) are a leading mammalian model of severe mitochondrial disease that exhibit many characteristic symptoms of Leigh Syndrome including oxidative stress, neuroinflammation, brain lesions, and premature death. NDUFS4 knockout mice have decreased expression of nearly every Complex I subunit. As Complex I normally contains at least 8 iron-sulfur clusters and more than 25 iron atoms, we asked whether a deficiency of Complex I may lead to iron perturbations, thereby accelerating disease progression. Consistent with this, iron supplementation accelerates symptoms of brain degeneration in these mice, while iron restriction delays the onset of these symptoms, reduces neuroinflammation, and increases survival. NDUFS4 knockout mice display signs of iron overload in the liver including increased expression of hepcidin and show changes in iron-responsive element-regulated proteins consistent with increased cellular iron that were prevented by iron restriction. These results suggest that perturbed iron homeostasis may contribute to pathology in Leigh Syndrome and possibly other mitochondrial disorders.

## Editor's evaluation

This is an important study showing that the perturbation of NDUFS4, a component of mitochondrial respiratory complex I, which is also a key consumer of iron in the cell, can perturb iron metabolism, causing hepatic iron overload and neurological dysfunction. Compellingly, iron chelation reverses some of these pathological phenotypes. This paper will be of broad interest particularly to the neurology and iron biology communities, for its novel observations and experimental rigor.

## Introduction

Inherited mitochondrial defects cause several lethal mitochondriopathies such as Leigh Syndrome, MELAS, and Friedreich's Ataxia (*Stenton and Prokisch, 2020*). Reduced assembly and/or activity of respiratory Complex I or other complexes of the electron transport chain (ETC) are widely implicated in the etiology of most of these mitochondrial diseases (*Stenton and Prokisch, 2020*; *Wallace, 2005*; *Nunnari and Suomalainen, 2012*; *Chan, 2006*). Additionally, a common feature of age-related diseases, including Alzheimer's disease, Parkinson's disease (PD), heart disease, and diabetes, is decreased oxidative phosphorylation through aberrant ETC function (*Kaeberlein, 2017*; *Kaeberlein et al., 2015*; *Kennedy et al., 2014*; *López-Otín et al., 2013*; *Shigenaga et al., 1994*). Complex I is

**eLife digest** Iron is a mineral that contributes to many vital body functions. But as people age, it accumulates in many organs, including the liver and the brain. Excess iron accumulation is linked to age-related diseases like Parkinson's disease.

Too much iron may contribute to harmful chemical reactions in the body. Usually, the body has systems in place to mitigate this harm, but these mechanisms may fail as people age. Uncontrolled iron accumulation may damage essential proteins, DNA and fats in the brain. These changes may kill brain cells causing neurodegenerative diseases like Parkinson's disease.

Mitochondria, the cell's energy-producing factories, use and collect iron inside cells. As people age, mitochondria fail, which is also linked with age-related diseases. It has been unclear if mitochondrial failure may also contribute to iron accumulation and associated diseases like Parkinson's.

Kelly et al. show that mitochondrial dysfunction causes iron accumulation and contributes to neurodegeneration in mice. In the experiments, Kelly et al. used mice with a mutation in a key-iron processing protein in mitochondria. These mice develop neurodegenerative symptoms and die early in life. Feeding the mice a high-iron diet accelerated the animals' symptoms. But providing them with an iron-restricted diet slowed their symptoms and extended their lives. Low-iron diets also slowed iron accumulation in the animal's liver and reduced brain inflammation.

The experiments suggest that mitochondrial dysfunction contributes to both iron overload and brain degeneration. The next step for scientists is understanding the processes leading to mitochondrial dysfunction and iron accumulation. Then, scientists can determine if they can develop treatments targeting these processes. This research might lead to new treatments for Parkinson's disease or other age-related conditions caused by iron overload.

the largest of the ETC complexes and is made up of 45 subunit proteins that represent a considerable portion of the protein mass of the inner mitochondrial membrane (*Hirst, 2013*). It coordinates the transfer of electrons from NADH to ubiquinone via a shuttle of eight or more redox-active iron-sulfur clusters found on the peripheral arm in the mitochondrial matrix, but little is known on the effects of improper regulation, assembly, biogenesis, and dynamics of these iron-sulfur clusters in the pathophysiology of Complex I deficiencies.

One of the most prevalent hereditary mitochondrial diseases is Leigh Syndrome, which is characterized by lactic acidosis, neuroinflammation, brain lesions of the basal ganglia, and death within the first few years of life (*Darin et al., 2001*). Nearly 35% of Leigh Syndrome cases can be caused by various mutations affecting Complex I including NDUFS4 and several other iron-sulfur proteins on the redox-active peripheral arm (*Stenton and Prokisch, 2020*; *Chang et al., 2020*). Mice missing the Complex I subunit NDUFS4 are a leading mammalian model of Leigh Syndrome (*Kruse et al., 2008*). We previously reported knockout of the iron-sulfur protein NDUFS4 ($Ndufs4^{-/-}$) in mice decreases the expression of nearly all Complex I subunits, and we were unable to observe appreciable Complex I or respiratory supercomplex formation (*Martin-Perez et al., 2020*; *Johnson et al., 2013*). Iron-sulfur cluster deficiencies in cells have recently been linked to an accumulation of iron (*Terzi et al., 2021*), and prior evidence in cells suggests inhibition of Complex I with rotenone promotes iron accumulation that is dependent on iron sensor protein activity (*Mena et al., 2011*; *Liang et al., 2020*; *Urrutia et al., 2017*; *Lee et al., 2009*). However, altered iron metabolism and its improper regulation because of deficiencies of an iron-sulfur cluster protein in $Ndufs4^{-/-}$ mice is unknown.

Iron, as an essential nutrient, is the most biologically abundant transition metal that plays key roles in physiology (*Hentze et al., 2010*; *Bleackley and Macgillivray, 2011*; *Grillo et al., 2017*). It is pervasively utilized as an enzymatic co-factor due to its unique readiness to undergo facile redox cycling in cellular milieu. Iron is largely localized to mitochondria due to its essential role in electron transfer during cellular respiration and in mitochondrial metabolism (*Pierrel et al., 2007*). The high reactivity of iron, however, facilitates reactive oxygen species generation, oxidative stress, ferroptosis, and organ damage when in excess (*Valko et al., 2005*; *Stohs and Bagchi, 1995*; *Lieu et al., 2001*). Iron homeostasis is tightly controlled through multiple transcriptional, translational, and post-translational iron-dependent feedback regulatory mechanisms such as the iron-responsive element (IRE) signaling pathway and the hepcidin-ferroportin axis (*Hentze et al., 2010*; *Bleackley and Macgillivray, 2011*).

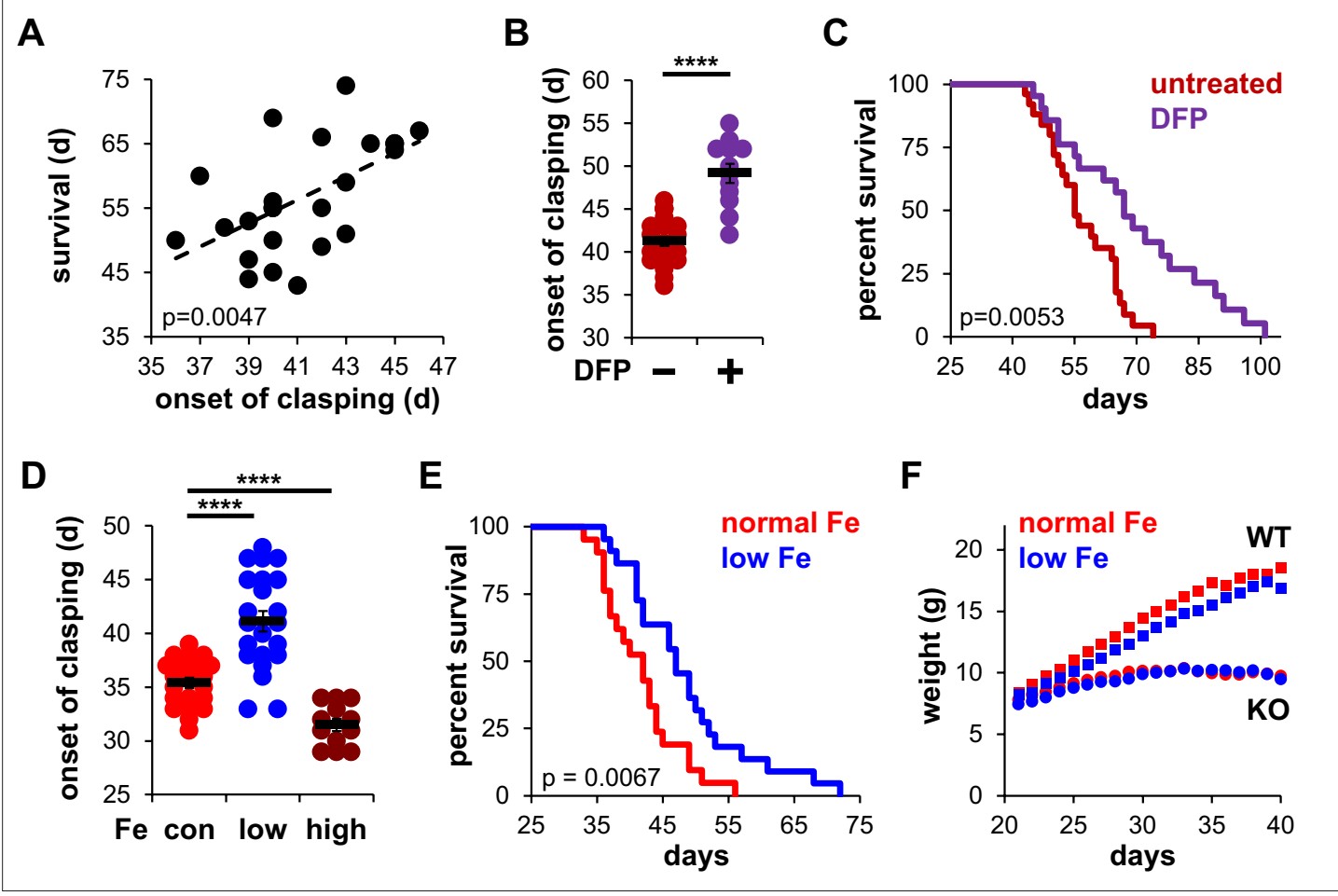

**Figure 1.** Iron restriction delays mitochondrial disease in mice. (**A**) Correlation between the onset of clasping and survival. Each point represents data from a single mouse. p=0.0047, Pearson's test. (**B**) Age at which *Ndufs4*$^{-/-}$ mice exhibited the clasping phenotype on chow diet. Mice were treated with either vehicle or deferiprone (DFP) in the water (2 mg/mL) from weaning. (**C**) Survival curves of *Ndufs4*$^{-/-}$ mice fed a chow diet and treated with deferiprone in the water (2 mg/mL) from weaning. (**D**) Onset of clasping in *Ndufs4*$^{-/-}$ mice on AIN-93G synthetic diet containing normal (40 ppm, con) or low (8 ppm) iron starting from weaning. Mice on control diet (40 ppm, Fe) were also treated with iron-dextran (100 mg/kg every 3 days via i.p. injection, high) from weaning. (**E**) Survival curves of mice on normal (40 ppm) or low (8 ppm) AIN-93G synthetic diet. (**F**) Weight gain in wild-type (WT, square markers) or *Ndufs4*$^{-/-}$ mice (KO, circle markers) on AIN-93G synthetic diet containing normal (40 ppm, red) or low (8 ppm, blue) concentrations of iron. p Value was calculated by log-rank for lifespan analyses. ****p<0.0001, t test with Bonferroni Correction.

The online version of this article includes the following figure supplement(s) for figure 1:

**Figure supplement 1.** Brain-permeable iron chelators are effective in delaying mitochondrial disease.

**Figure supplement 2.** Iron restriction induces iron-deficiency anemia.

However, abnormal iron accumulation and/or utilization can overwhelm these regulatory mechanisms leading to disease (*Bleackley and Macgillivray, 2011*). Here, we utilized the *Ndufs4*$^{-/-}$ mice to test the hypothesis that Complex I deficiencies may alter normal iron distribution which contributes to mitochondrial disease progression.

## Results
### Iron status underlies disease progression in NDUFS4-KO mice

To evaluate the influence of iron on disease progression in *Ndufs4*$^{-/-}$ mice, we first observed the onset of clasping in mice treated with the FDA-approved iron chelator deferiprone. Clasping is a common feature in the early stages of brain degeneration in these mice that immediately precedes severe neuroinflammation, weight loss, and ataxia (*Johnson et al., 2013*). There is a high correlation between

the onset of clasping and observed lifespan (*Figure 1A*), making this neurobehavioral symptom an appropriate readout of disease progression. Control *Ndufs4⁻/⁻* mice fed standard chow began clasping around 40 days of age, consistent with our prior reports (*Martin-Perez et al., 2020*). To probe the role of iron in neurodegeneration, we added the FDA-approved iron chelator deferiprone to the water of *Ndufs4⁻/⁻* mice after weaning. We observed treatment with this brain-penetrating iron chelator delayed the onset of clasping (*Figure 1B* and *Figure 1—figure supplement 1A*). Deferiprone treatment also increased median lifespan in *Ndufs4⁻/⁻* mice (*Figure 1C* and *Figure 1—figure supplement 1B and C*). The hydrophilic iron chelator deferoxamine does not readily cross the blood-brain barrier (*Liu et al., 2005*) and was used to ask whether brain iron chelation was necessary for the observed effects. In contrast to deferiprone, we observed no changes upon deferoxamine treatment on clasping or lifespan (*Figure 1—figure supplement 1A and D*). This suggests that local iron chelation in the brain is critical for the observed effects.

Because robust quantification of iron clearance is difficult to achieve due to an unreliable variability of iron in non-synthetic chows, we performed similar experiments using a synthetic AIN-93G control diet containing normal levels of iron (40 ppm). It was reported that *Ndufs4⁻/⁻* mice fed a synthetic diet clasp at earlier ages and have shorter lifespans than *Ndufs4⁻/⁻* mice fed standard chow diets (*Grillo et al., 2021*), which we similarly observed (*Figure 1D and E*). High iron supplementation via intraperitoneal injection of iron-dextran (100 mg/kg every 3 days) accelerated the onset of clasping (*Figure 1D*), while feeding mice a low iron (8 ppm) synthetic diet dramatically delayed the onset of clasping without causing any significant deleterious changes in weight (*Figure 1D and F*). This delay in disease progression was associated with ~15% increase in lifespan (*Figure 1E*).

We next performed hematological analysis of blood isolated from these mice. As expected, *Ndufs4⁻/⁻* mice fed a low iron diet showed signs of microcytic, hypochromic anemia consistent with iron deficiency. We observed decreased hematocrit, reduced hemoglobin levels, and decreased red

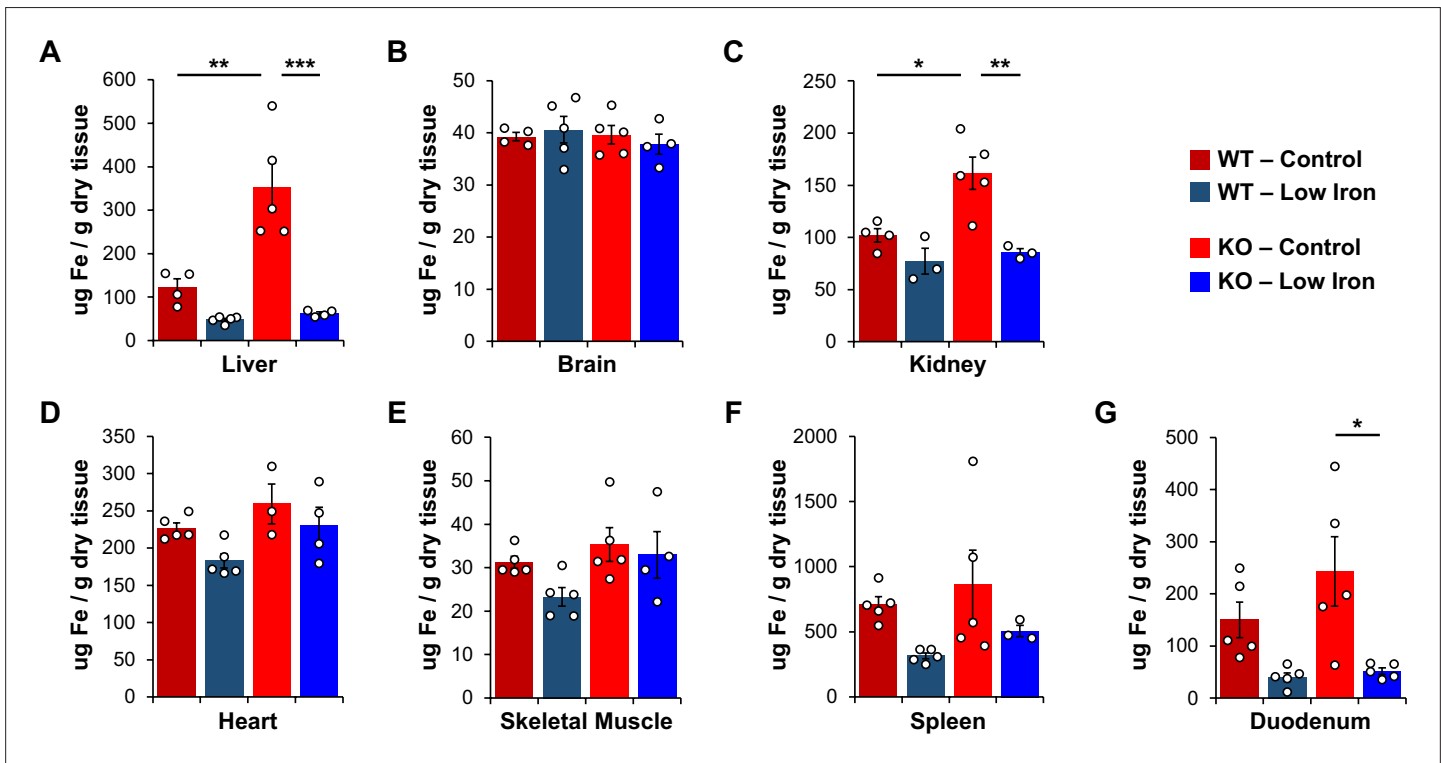

**Figure 2.** Total iron quantification in tissues. Quantification of total iron by ICP-MS from WT and *Ndufs4⁻/⁻* mice at PND35 fed control (40 ppm) or low (8 ppm) AIN-93G in (**A**) liver, (**B**) whole brain, (**C**) kidney, (**D**) heart, (**E**) quadricep, (**F**) spleen, and (**G**) duodenum. N=3–5 mice. *p<0.05, **p<0.01, ***p<0.001, ANOVA with post hoc Tukey. ICP-MS, inductively coupled plasma mass spectrometry; WT, wild-type.

The online version of this article includes the following figure supplement(s) for figure 2:

**Figure supplement 1.** Analysis of biologically-relevant transition metals.

**Figure supplement 2.** Iron restriction has no effect on organ size.

blood cell production in *Ndufs4⁻/⁻* mice fed a low iron diet compared to the normal iron diet cohorts (*Figure 1—figure supplement 2A–C*). We also observed decreases in mean corpuscular volume, mean corpuscular hemoglobin, and mean corpuscular hemoglobin concentration with the low iron diet (*Figure 1—figure supplement 2D–F*).

## Metal imbalances in tissues of NDUFS4-KO mice

We collected liver and other tissues from wild-type (WT) and *Ndufs4⁻/⁻* mice near the onset of clasping (i.e., PND35) to quantify total iron levels by inductively coupled plasma mass spectrometry (ICP-MS).

**Table 1.** ICP-MS quantification of biologically relevant metals in WT and *Ndufs4⁻/⁻* tissues at PND35 fed a normal (40 ppm) and low (8 ppm) iron AIN-93G synthetic diet.

Metals were measured as µg metal relative to total dry weight of tissue. N=3–5 mice, - $p<0.10$, *$p<0.05$, **$p<0.01$, ***$p<0.001$, ****$p<0.0001$, ANOVA with post hoc Tukey.

| Tissue | Metal (µg/g) | *Ndufs4⁺/⁺* Mice AIN-93G normal iron | *Ndufs4⁺/⁺* Mice AIN-93G low iron | *Ndufs4⁻/⁻* Mice AIN-93G normal iron | *Ndufs4⁻/⁻* Mice AIN-93G low iron | p Value WT-Con versus KO-Con | p Value KO-Con versus KO-Low |
|---|---|---|---|---|---|---|---|
| | Fe | 122.9±18.9 | 48.0±3.5 | 352.5±55.5 | 62.5±3.6 | ** | *** |
| | Mn | 2.20±0.33 | 5.68±0.63 | 3.70±0.44 | 8.62±0.51 | | **** |
| | Zn | 47.5±5.07 | 62.0±5.8 | 60.9±5.0 | 71.2±4.0 | | |
| Liver | Cu | 9.6±0.64 | 12.7±0.9 | 13.9±1.5 | 15.6±1.2 | - | |
| | Fe | 39.3±0.78 | 40.6±2.5 | 39.6±1.8 | 37.8±1.9 | | |
| | Mn | 1.46±0.04 | 1.94±0.05 | 1.71±0.09 | 1.91±0.09 | | |
| | Zn | 37.7±0.53 | 48.3±1.6 | 44.1±1.8 | 49.1±3.1 | | |
| Brain | Cu | 10.3±0.52 | 11.4±0.5 | 11.8±0.4 | 12.1±0.7 | | |
| | Fe | 101.9±6.4 | 77.1±12.4 | 161.5±15.4 | 85.8±3.5 | * | ** |
| | Mn | 4.54±0.44 | 5.53±0.31 | 4.48±0.20 | 6.01±0.51 | | * |
| | Zn | 55.8±1.8 | 56.6±2.6 | 58.9±4.8 | 58.4±5.7 | | |
| Kidney | Cu | 15.1±0.3 | 15.3±0.9 | 17.5±0.7 | 18.0±0.9 | - | |
| | Fe | 226.8±6.9 | 182.7±9.5 | 259.2±26.9 | 230.7±24.0 | | |
| | Mn | 2.51±0.09 | 3.00±0.05 | 2.59±0.19 | 3.72±0.35 | | ** |
| | Zn | 46.6±3.0 | 47.7±1.5 | 31.5±3.0 | 41.3±5.3 | * | |
| Heart | Cu | 30.5±1.2 | 30.5±0.1 | 31.7±2.9 | 40.4±2.8 | | * |
| | Fe | 31.2±1.4 | 23.3±2.1 | 35.4±3.9 | 33.0±5.3 | | |
| | Mn | 0.53±0.05 | 0.83±0.08 | 0.65±0.08 | 0.88±0.10 | | |
| | Zn | 20.7±0.7 | 21.8±1.4 | 22.5±2.8 | 18.8±1.8 | | |
| Skeletal Muscle | Cu | 3.80±0.24 | 3.85±0.22 | 4.60±0.56 | 4.40±0.44 | | |
| | Fe | 710.4±59.1 | 316.9±22.3 | 860.5±265.6 | 507.1±43.7 | | |
| | Mn | 0.94±0.09 | 1.28±0.08 | 0.92±0.40 | 1.45±0.23 | | |
| | Zn | 110.5±7.5 | 123.1±10.4 | 109.0±38.7 | 126.0±6.8 | | |
| Spleen | Cu | 5.13±0.55 | 6.43±0.43 | 3.84±1.26 | 5.29±0.57 | | |
| | Fe | 150.1±34.2 | 40.0±8.6 | 243.0±66.3 | 51.7±6.2 | | * |
| | Mn | 6.78±1.2 | 12.88±2.63 | 6.76±1.28 | 17.1±1.7 | | ** |
| | Zn | 89.2±2.2 | 84.9±12.7 | 101.4±9.6 | 114.1±9.3 | | |
| Duodenum | Cu | 9.4±0.3 | 8.4±1.3 | 10.8±0.9 | 11.2±0.8 | | |

This time point was chosen as it is near the median age that control $Ndufs4^{-/-}$ mice begin clasping (*Figure 1D*). ICP-MS of tissue digestates revealed that total liver iron levels tripled in $Ndufs4^{-/-}$ mice fed the control synthetic diet relative to WT mice (*Figure 2A* and *Table 1*). Significantly decreased iron levels in kidney and duodenum were observed in mice fed the low iron diet (*Figure 2A–G* and *Table 1*), and we observed a global reduction in iron levels in the tissues tested (*Figure 2—figure supplement 1A*). Iron restriction did not affect the relative weights of these tissues compared to mice fed the control diet (*Figure 2—figure supplement 2*). We observed no differences in total iron in whole brain digestates from 35-day-old WT and $Ndufs4^{-/-}$ mice fed the normal or low iron diets (*Figure 2B* and *Table 1*).

Similarities between divalent metals, notably iron (II) and manganese (II), lead to low stoichiometric selectivity of divalent metal transport proteins for these metals (*Ye et al., 2017*). These transport proteins often have higher affinities for copper or manganese. The biological selectivity, however, is primarily controlled by the relative availability of these metals in the cell (*Grillo et al., 2017*; *Finney and O'Halloran, 2003*; *Cyert and Philpott, 2013*; *Ba et al., 2009*). Labile iron is most often in ~10-fold or greater concentrations relative to manganese, copper, or other metals in biological systems, and thus iron is often preferred. In cases of iron restriction, however, the observed selectivity for transport through metal transport proteins shifts to prefer manganese. Manganese metabolism is sensitive to changes in iron homeostasis because of the promiscuity of many iron transport proteins for other divalent metals (*Ye et al., 2017*).

We thus quantified total levels of manganese and other metals in WT and $Ndufs4^{-/-}$ mice on normal or low iron diet by ICP-MS. Consistent with the putative effect that iron restriction has on increasing manganese uptake, we observed elevated manganese levels in WT and $Ndufs4^{-/-}$ mice fed a low iron diet in the liver, brain, kidney, and other tissues relative to control mice (*Figure 2—figure supplement 1B* and *Table 1*). The control $Ndufs4^{-/-}$ mice had increased basal manganese levels compared to WT cohorts. This may be due to a compensatory upregulation of mitochondrial manganese-dependent superoxide dismutase (MnSOD aka SOD2) to combat iron-mediated oxidative stress when mice were fed the normal iron diet (*Lee et al., 2019*). We did not observe any notable changes in copper, zinc, or other biorelevant metals in iron-restricted mice (*Table 1* and *Figure 2—figure supplement 1C and D*). However, $Ndufs4^{-/-}$ mice fed the normal iron diet showed increased copper levels relative to control WT mice (*Figure 2—figure supplement 1D*).

## Increased oxidative stress and neuroinflammation in NDUFS4-KO mice

Excess free iron often accelerates the accumulation of reactive oxygen species by reacting with molecular oxygen through the Fenton reaction. $Ndufs4^{-/-}$ mice have been reported to have elevated, and presumably toxic, levels of oxygen in the brain, blood, and other tissues due to their decreased respiratory activity (*Jain et al., 2019*). The unmitigated accumulation of reactive oxygen species causes oxidative stress and damages cells through the radical-mediated peroxidation of polyunsaturated fatty acids (PUFAs). These lipid peroxyl radicals break down into malondialdehyde (MDA) and 4-hydroxynonenal and promote local inflammation.

We first asked whether $Ndufs4^{-/-}$ mice on control iron diet had increased non-heme iron relative to WT controls using a ferrozine assay. Ferrozine forms a purple-colored chelate with iron allowing for the colorimetric detection and quantification of free or weakly bound (e.g., non-heme) iron. This free iron can react with $O_2$ and PUFAs to generate ROS. Consistent with this, we observed livers from $Ndufs4^{-/-}$ mice on the control iron diet had increased non-heme iron (*Figure 3A*).

This led us to next ask whether livers from $Ndufs4^{-/-}$ mice had increased PUFA oxidation using MDA as a readout. We quantified MDA levels using a TBARS assay, in which thiobarbituric acid reacts with MDA to form a TBA-MDA adduct that can be quantified colorimetrically. We observed a non-significant trend toward higher MDA levels in livers from control $Ndufs4^{-/-}$ mice compared to WT cohorts (*Figure 3B*). Noting that this time point (P35) represents the median age when mice begin to show symptoms of disease, we asked whether the $Ndufs4^{-/-}$ mice that already show symptoms have elevated MDA levels. We observed a strong correlation between the length of time $Ndufs4^{-/-}$ mice exhibited neurodegenerative symptoms (i.e., clasping) and observed MDA levels (*Figure 3C*). Iron-deficient mice showed decreased MDA levels (*Figure 3B*). Collectively, this data support the hypothesis that iron accumulation in livers of $Ndufs4^{-/-}$ mice promotes oxidative stress.

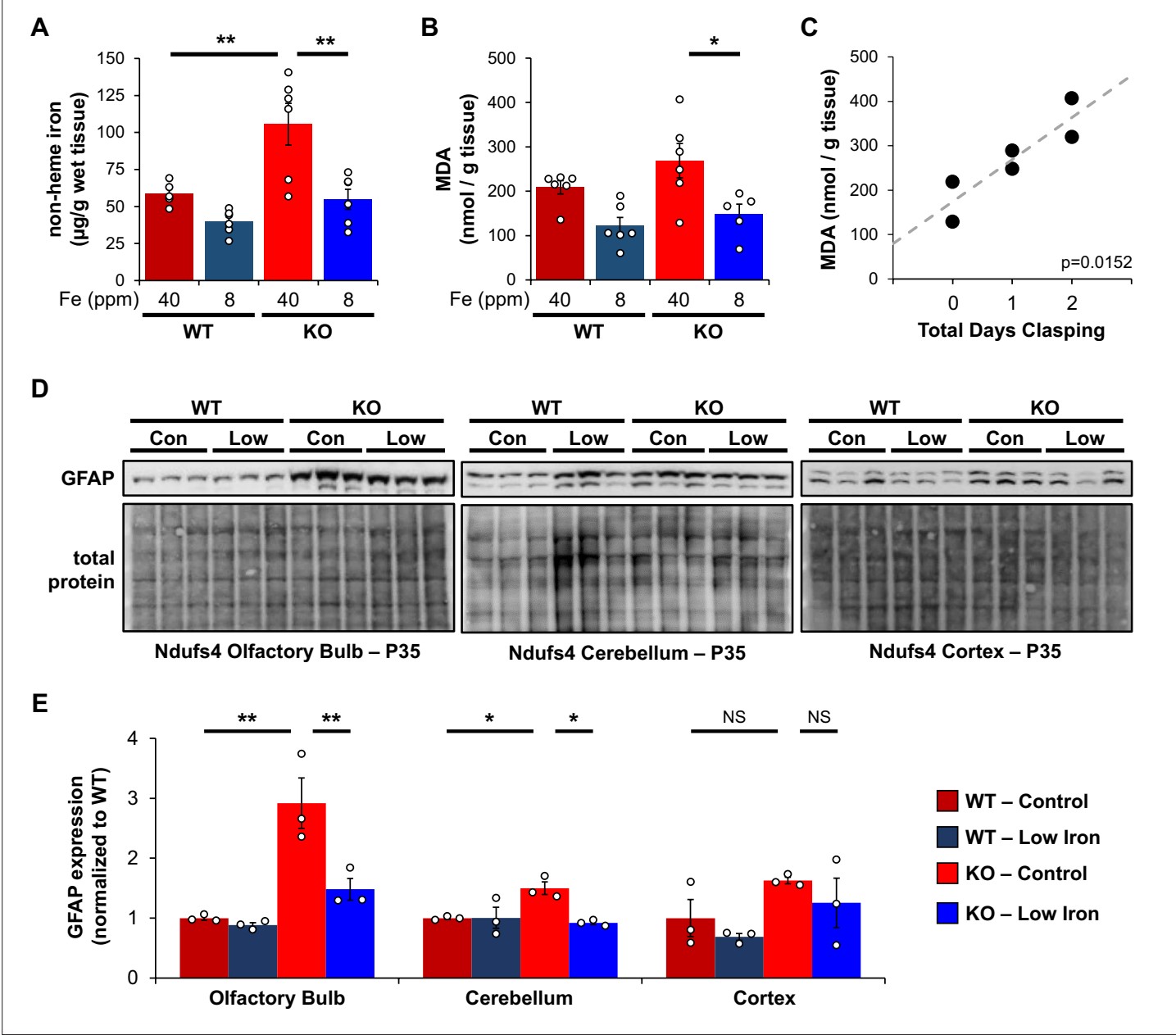

**Figure 3.** Iron restriction reduces iron-dependent oxidative damage and neuroinflammation. (**A**) Quantification of non-heme iron by ferrozine assay and (**B**) MDA-TBA adduct in livers from WT and *Ndufs4⁻/⁻* mice at PND35 that were fed control (40 ppm) or low (8 ppm) AIN-93G synthetic diet. (**C**) Correlation between days since *Ndufs4⁻/⁻* mice began displaying the clasping phenotype with detected liver MDA levels from (**B**). p=0.0152, Pearson's test. (**D**) Representative western blot images and (**E**) densitometry (relative to total protein) of the astrogliosis marker GFAP from brain sections that normally exhibit brain lesions (olfactory bulb, cerebellum) and that do not (cortex) from PND35 WT and *Ndufs4⁻/⁻* mice fed a control (40 ppm) or low (8 ppm) AIN-93G synthetic diet. Each lane represents protein extract from a single mouse. N=3–6 mice. *p<0.05, **p<0.01, ANOVA with post hoc Tukey. MDA-TBA, malondialdehyde thiobarbituric acid; WT, wild-type.

The online version of this article includes the following source data for figure 3:

**Source data 1.** Raw unedited immunoblots for *Figure 3D* brain regions.

*Ndufs4⁻/⁻* mice normally exhibit neuroinflammation and brain lesions in the olfactory bulb, cerebellum, and brain stem around the onset of clasping, while other brain regions are largely unaffected (***Johnson et al., 2020***). We did not observe significant differences in the astrogliosis marker GFAP in whole brain extracts. However, we observed increased GFAP expression in the olfactory bulb and cerebellum of *Ndufs4⁻/⁻* mice fed the control iron diet while GFAP levels in the cortex were unchanged

(*Figure 3D and E*). Consistent with the capacity for iron restriction to delay clasping (*Figure 1D*), GFAP levels were reduced in these regions in iron-restricted mice (*Figure 3D and E*).

## IRE iron regulation suggests increased labile iron

Increased MDA production suggests increased labile iron, but it is challenging to directly probe changes in intracellular iron distribution in vivo. However, the expression of iron-dependent regulatory proteins can be used as an effective readout. Body iron distribution is solely controlled by regulating iron absorption, metabolism, and storage as there are no known adaptive biological mechanisms for iron excretion (*Lieu et al., 2001*). The expression of iron regulatory proteins (IRPs) is highly sensitive to intracellular iron perturbations. For example, upregulation of the iron storage protein ferritin helps reduce cellular labile iron and avoid oxidative stress in cases of iron overload. Regulation is primarily achieved at the translational and post-translational levels through IREs and the hepcidin-ferroportin axis (*Hentze et al., 2010*; *Bleackley and Macgillivray, 2011*).

IREs are found in the 5' or 3'-untranslated regions of mRNA coding for proteins involved in iron uptake (e.g., transferrin receptor 1), iron storage (e.g., ferritin), and cellular iron export (e.g., ferroportin). The IRPs bind to these IREs and either block translation (5'-IRE) or prevent endonuclease-mediated mRNA degradation (3'-IRE) upstream of the poly-A tail (*Figure 4—figure supplement 1*). Recognition of labile iron by IRPs induces a conformational change that prevents IRPs binding to the IREs. Holo-IRP1 adopts aconitase activity in the cytosol while IRP2 is ubiquitinated and degraded in cases of iron overload.

To probe changes in cellular iron status, we evaluated mRNA levels and expression of proteins known to be involved in IRE-dependent regulation. Expression of the cytosolic iron storage protein ferritin is controlled by 5'-IREs found on its 24 heavy (*Fth1*) and light (*Ftl1*) chain subunits and is an excellent readout of free cellular iron. Ferritin can deposit more than 4000 iron atoms into its central core, thereby acting as an iron sponge (*Harrison and Arosio, 1996*). In cases of iron overload, IRPs dissociate from the 5'-IRE allowing for translation of FTH1 and FTL1 (*Figure 4—figure supplement 1*). Consistent with iron overload in livers of control $Ndufs4^{-/-}$ mice, we observed knockout of NDUFS4 increased FTH1 protein expression (*Figure 4*). Iron restriction in $Ndufs4^{-/-}$ mice downregulated FTH1 expression, consistent with iron deficiency anemia (*Figure 4*).

Transferrin Receptor 1 (TFR1) mediates the cellular internalization of iron-bound transferrin. *Tfr1* mRNA contains a 3'-IRE and is normally expressed at high levels with iron restriction. To adapt to iron overload, IRPs dissociate from the *Tfr1* mRNA leading to its endonuclease-mediated degradation (*Figure 4—figure supplement 1*). Control $Ndufs4^{-/-}$ mice showed decreased levels of *Tfr1* mRNA in liver by qRT-PCR relative to WT mice (*Figure 5A*), consistent with increased labile iron in the mitochondrial disease mice. As expected, we observed upregulation of *Tfr1* mRNA and TFR1 protein in livers from the WT and $Ndufs4^{-/-}$ mice fed the low iron diet (*Figure 4* and *Figure 5A*). We also observed changes in other IRE-regulated genes at the mRNA or protein level in liver as expected for IRE-dependent regulation (*Figure 4* and *Figure 5A–E*). Iron deficiency in $Ndufs4^{-/-}$ mice prevented the altered expression of these genes (*Figure 5A–E*), consistent with reduced cellular iron. We did not observe appreciable differences in FTH1 or TFR1 expression in the brain (*Figure 4—figure supplement 2*).

## The hepcidin-ferroportin axis in a Complex I deficiency

The peptide hepcidin is a liver-derived master regulator of iron absorption, iron recycling, iron storage, and in erythropoiesis (*Hentze et al., 2010*). In situations of excess iron, hepatic production and systemic circulation of hepcidin promote ubiquitination and proteasomal degradation of the iron export protein ferroportin (FPN1), thereby trapping iron in duodenal epithelia and in liver macrophages. This promotes the storage of excess iron in the liver and reduces iron absorption from the diet as an effective adaptive mechanism protecting against iron-mediated damage in cases of iron overload. While *Fpn1* mRNA contains a 5'-IRE, hepcidin-mediated degradation of FPN1 prevents its IRE-dependent upregulation in the liver or gut epithelia with iron overload. Consistent with this, we did not observe significant increases in FPN1 protein in $Ndufs4^{-/-}$ liver (*Figure 4*).

We thus asked whether hepcidin production increased in livers of $Ndufs4^{-/-}$ mice. We quantified *Hamp1* mRNA encoding the hepcidin peptide (*Figure 5F*). Transcript levels of *Hamp1* increased twofold in livers of control $Ndufs4^{-/-}$ mice, further consistent with our ferritin data suggesting increased

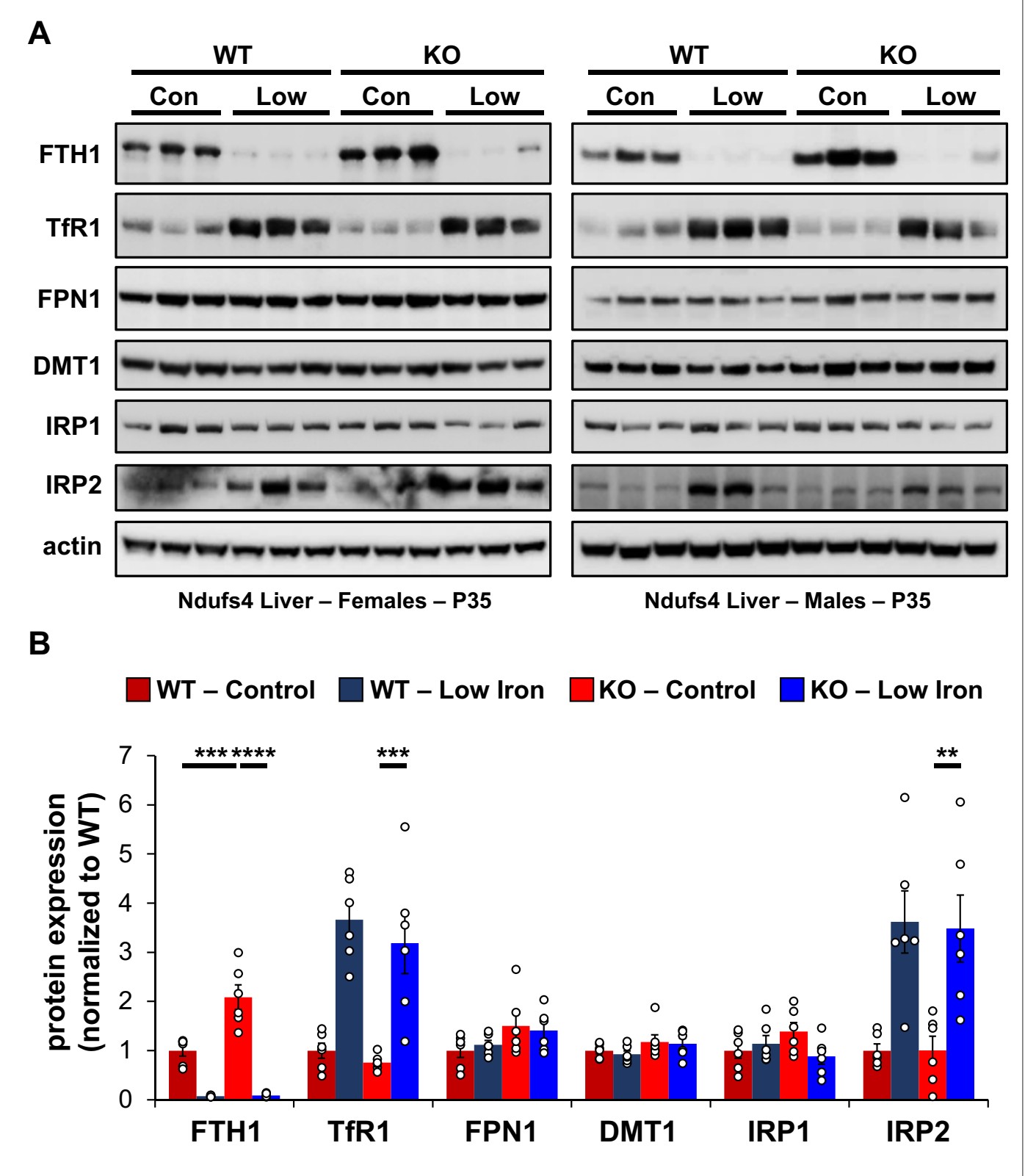

**Figure 4.** Changes in iron-dependent proteins suggest increased labile iron. (**A**) Representative western blot images and (**B**) densitometry (relative to actin) of proteins involved in regulation of iron transport, storage, or metabolism in livers from PND35 WT and *Ndufs4*⁻/⁻ mice fed a control (40 ppm) or low (8 ppm) iron AIN-93G synthetic diet from weaning. Each lane represents protein extract from a single mouse. \*\*p<0.01, \*\*\*p<0.001, \*\*\*\*p<0.0001, ANOVA with post hoc Tukey.

*Figure 4 continued on next page*

*Figure 4 continued*

The online version of this article includes the following source data and figure supplement(s) for figure 4:

**Source data 1.** Raw unedited immunoblots for *Figure 4A* liver.

**Figure supplement 1.** Simplified schematic of IRE-dependent translational regulation of proteins involved in iron transport, storage, or metabolism.

**Figure supplement 2.** Immunoblot of iron-dependent proteins in whole brain extracts.

**Figure supplement 2—source data 1.** Raw unedited immunoblot images for *Figure 4—figure supplement 2A* brain.

hepatic iron stores. Iron restriction drastically downregulated *Hamp1* transcription in liver to nearly undetectable levels (*Figure 5F*).

## Discussion

### Pathogenic iron dyshomeostasis in diverse age-related and genetic diseases

Iron is widely recognized as a critical mediator in the progression of many age-related and genetic neurodegenerative diseases (*Lieu et al., 2001*). There is increased interest in recent years to better

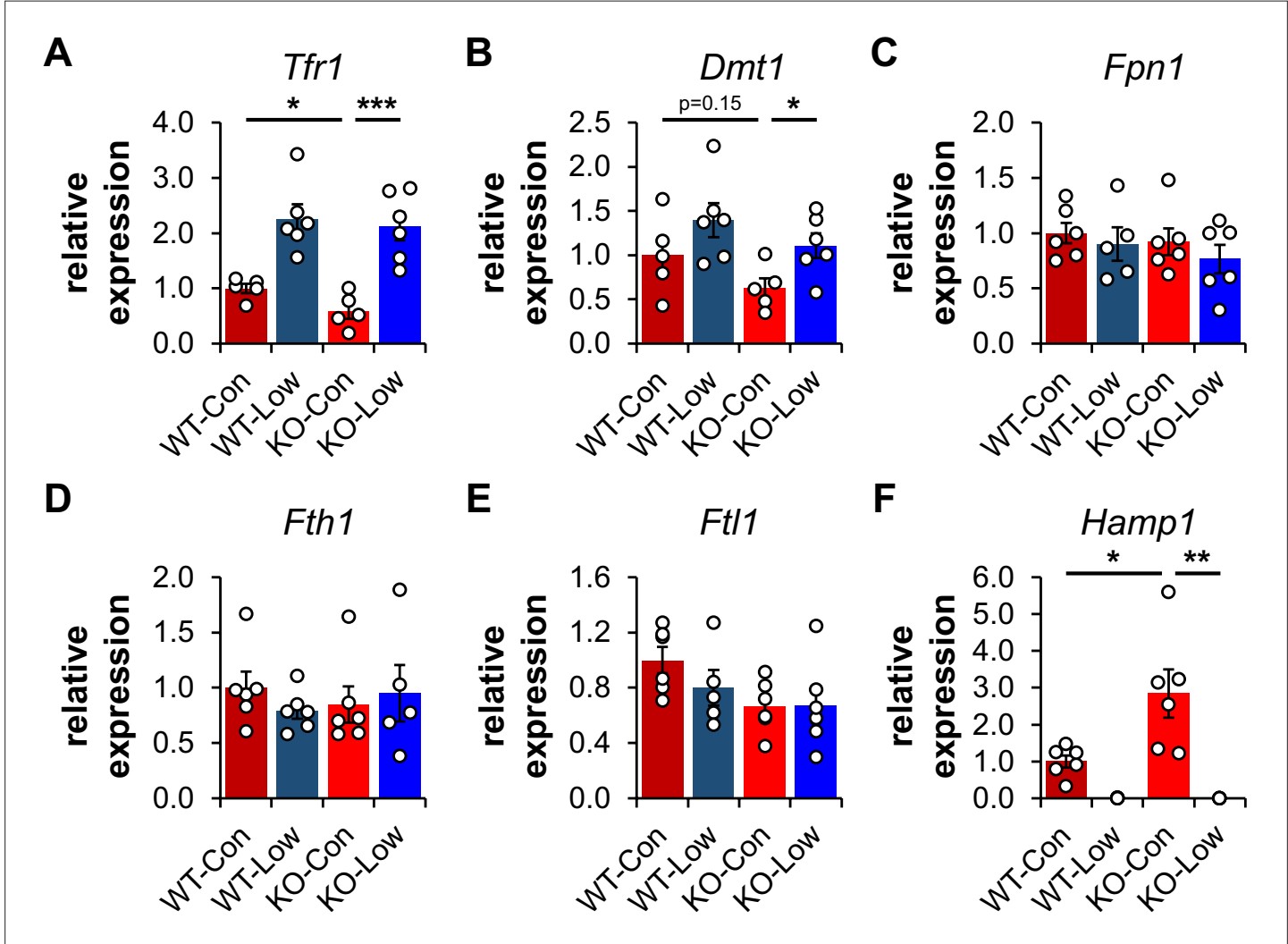

**Figure 5.** Expression profiling of IRE-containing genes by qPCR. (**A**) Quantification of relative mRNA expression of *Tfr1*, (**B**) *Dmt1*, (**C**) *Fpn1*, (**D**) *Fth1* (ferritin heavy chain 1), (**E**) *Ftl1* (ferritin light chain 1), and (**F**) *Hamp* (hepcidin) in livers from PND35 WT and *Ndufs4$^{-/-}$* (KO) mice fed a control (40 ppm) and low (8 ppm) iron AIN-93G synthetic diet from weaning. *p<0.05, **p<0.01, ***p<0.001, t test with Bonferroni correction.

understand its pathogenic role in neurodegeneration with brain iron accumulation and other age-related disorders (*Zecca et al., 2004*). For example, iron deposition in the basal ganglia of PD patients and mouse models accelerates the aggregation of alpha-synuclein, promotes free radical accumulation, increases lipid peroxidation, and is implicated in dementia and motor impairment (*Zecca et al., 2004*). Treatment with the Complex I inhibitor rotenone is widely used as a PD model that reproduces many features including oxidative damage, dopaminergic degeneration in the substantia nigra, and Lewy Body inclusions (*Liang et al., 2020*). Rotenone treatment increases the mitochondrial and cytosolic labile iron pool in neurons in vitro and in animal models (*Mena et al., 2011*; *Liang et al., 2020*; *Urrutia et al., 2017*; *Lee et al., 2009*).

Several mitochondrial disorders caused by deficiencies of other ETC complexes or assembly factors can result in clinical symptoms overlapping with Leigh Syndrome. For example, point mutations of the Complex III chaperone ubiquinol-cytochrome c reductase complex chaperone (BCS1L) can cause the rare hereditary disease GRACILE (Growth Retardation, Aminoaciduria, Cholestasis, Iron overload, Lactic acidosis, and Early death) syndrome (*Visapää et al., 2002*; *Fellman et al., 1998*; *Fellman, 2002*; *Fellman et al., 2008*; *Levéen et al., 2011*). BCS1L normally mediates the transfer of Rieske iron-sulfur protein into the pre-assembled Complex III dimer, but its deficiency leads to elevated liver iron, increased serum iron or ferritin, and hypotransferrinemia (*Visapää et al., 2002*). There is considerable overlap between the metabolic alterations in patients with Leigh Syndrome and GRACILE syndrome. This is demonstrated by findings that some mutations of BCS1L cause Leigh Syndrome (e.g., P99L) (*de Lonlay et al., 2001*), while other mutations in this protein alternatively cause GRACILE syndrome (e.g., S78G) (*Fellman et al., 2008*). Similar metabolic changes include decreased respiratory supercomplex formation, reduced $O_2$ consumption, high amino acid levels, $NAD^+$ dyshomeostasis, increased glycolysis, and increased lactate fermentation (*Visapää et al., 2002*; *Fellman et al., 1998*; *Fellman, 2002*; *Fellman et al., 2008*; *Levéen et al., 2011*). As seen in Leigh Syndrome, GRACILE Syndrome patients with encephalopathy experience seizures, developmental delays, and other neurodegenerative symptoms (*de Lonlay et al., 2001*; *Morris et al., 1995*). These shared phenotypic manifestations of Leigh and GRACILE Syndrome may be a result of overlapping pathophysiology of iron accumulation.

## Iron accumulation in various tissues

Our ICP-MS data quantifying metals in several tissues strongly suggest that there is an accumulation of iron in the *Ndufs4⁻/⁻* mice (*Figure 2* and *Table 1*). Interestingly, we see the most drastic changes in the liver. The liver plays a central role in iron absorption, metabolism, and regulation by acting as the primary iron storage vessel to protect sensitive organs, such as the brain. The liver promotes the controlled release of iron into the blood to maintain non-toxic physiological levels, is largely responsible for synthesizing circulating iron-binding proteins such as transferrin, and produces hepcidin—the master regulating peptide of iron absorption—in response to changes in hepatic iron stores (*Hentze et al., 2010*; *Lieu et al., 2001*). Because of the liver's critical importance in maintaining systemic iron, our results suggest there is an excess of body iron in the setting of this Complex I deficiency. While the *Ndufs4⁻/⁻* mice experience a severe neurometabolic disease, we previously reported that liver-specific knockout of S6K1, but not neuronal S6K1, can delay the onset of disease and prolong survival (*Ito et al., 2017*). Our data further implicate aberrations in liver physiology in *Ndufs4⁻/⁻* mouse disease progression that may translate to patients with Leigh Syndrome.

We were unable to detect any noticeable differences in iron in whole brain digestates by ICP-MS (*Figure 2B*). The reason for this is unclear, however, it may be due to well-controlled brain-specific adaptive mechanisms that limit iron absorption through the blood-brain barrier (*Mills et al., 2010*), or due to differential iron distribution in regions that are unaffected in *Ndufs4⁻/⁻* mice (*Johnson et al., 2020*). For example, mice missing the iron regulatory protein IRP2 show neurocognitive deficits consistent with brain iron overload, but iron is only deposited in glial cells and white matter while neuronal iron contrastingly decreases (*Zumbrennen-Bullough et al., 2014*; *Ghosh et al., 2015*). Age-progressive iron accumulation is primarily found in the basal ganglia and other regions that control motor function while whole brain iron deposition is largely unchanged (*Ward et al., 2014*). In support of this, *Ndufs4⁻/⁻* pathology is commonly observed in the basal ganglia, cerebellum, and olfactory bulbs. These brain regions tend to have higher iron levels than other regions (*Zecca et al., 2004*; *Ward et al., 2014*). Iron does not need to cross the blood-brain barrier to accumulate in the olfactory bulb or brain stem, and as such these regions are more sensitive to disturbances in iron homeostasis.

We observed that dietary iron restriction, and treatment with an iron chelator, both delay the onset of neurological symptoms (i.e., clasping). This is consistent with the idea that altered whole body iron status can influence neurodegeneration. However, it is unclear whether changes in organ function upon iron perturbation contribute to the observed effects.

## Cellular iron perturbations as a result of a Complex I deficiency

Our data do not elucidate the specific forms or oxidation state of cellular iron that accumulates in $Ndufs4^{-/-}$ mice, but it is possible that the speciation of iron is altered. As Complex I normally utilizes >25 iron atoms in its iron-sulfur clusters, its deficiency could alter the iron composition in the cell. In support of this, recent research shows iron-sulfur cluster deficiencies cause iron imbalances (*Terzi et al., 2021*). In contrast to protein-bound iron, excess cellular labile iron is widely associated with ROS generation, lipid peroxidation, oxidative stress, ferroptosis, or other damaging processes (*Valko et al., 2005*). We observed increased iron (*Figure 3A*), increased levels of the lipid peroxidation byproduct MDA (*Figure 3C*), and changes in the IRE-associated proteins TFR1 and ferritin subunits (*Figure 4*). This data collectively support increased free cytosolic iron in $Ndufs4^{-/-}$ mouse liver that is recognized by IRP1 and IRP2.

Iron is especially localized to mitochondria due to its widespread use in energy production, metabolism, and normal ROS signaling. A large proportion of this is normally utilized in oxidative phosphorylation. It is thus interesting to consider whether the distribution of iron is perturbed not only at the organ level, but also intracellularly, in $Ndufs4^{-/-}$ mice. A recent report in rotenone-treated SH-SY5Y neuroblastoma cells showed Complex I inhibition increased both cytosolic and mitochondrial labile iron by calcein-green and RPA fluorescence, respectively (*Liang et al., 2020*). As NDUFS4 is found in mitochondria, alterations in subcellular iron distribution remain an attractive avenue of pursuit in the setting of a Complex I deficiency.

## The effect of iron deficiency on oxygen status

Recent reports provide some evidence that $Ndufs4^{-/-}$ mice are hyperoxic, show increased $PO_2$ levels in the brain and blood, and have perturbed redox status (*Jain et al., 2019*). This excess of oxygen is implicated in disease etiology. Consistent with this, housing mice in low oxygen atmospheres (11% $O_2$) corrects abnormal gas tensions, decreases mitochondrial $H_2O_2$ production, prevents brain lesions, and considerably extends lifespan (*Ferrari et al., 2017*; *Jain et al., 2016*). Induction of severe anemia by regular early life phlebotomy every other day in combination with an iron-deficient diet is similarly beneficial (*Jain et al., 2019*). However, the content of this iron-restricted diet, the characterization of iron status, and the influence of iron on disease progression in $Ndufs4^{-/-}$ mice were unclear. Our data do not indicate whether the low iron diet decreases oxygen tension to delay the onset of neurodegenerative symptoms, but it may play a role. However, the extent of lifespan extension with a low iron diet is small in comparison to severe anemia by combining iron restriction with regular phlebotomy. Both may be necessary to drastically reduce oxygen delivery to the brain.

It was previously reported that genetic activation of the hypoxia-inducible factor (HIF) regulatory pathway via knockout of prolyl hydroxylases, nestin, or Von Hippel-Lindau protein in $Ndufs4^{-/-}$ mice is detrimental (*Jain et al., 2019*). Jain and Zazzeron et al. reported median lifespan in control $Ndufs4^{-/-}$ mice decreases from 64 to 29 days in $Ndufs4^{-/-}$ $Nes-Phd2^{-/-}$ triple transgenic mice that display HIF activation (*Jain et al., 2019*). Increased iron absorption via hyperactivation of hypoxia-inducible factors may partially explain this surprising result.

## Conclusion

This work establishes that mice missing the iron-sulfur cluster containing ETC subunit NDUFS4 have iron overload in the liver, and that iron restriction is sufficient to delay the onset of disease progression. Proteins sensitive to changes in cellular iron, such as TFR1 and ferritin, change in response to this iron overload. This is consistent with an increase in labile iron that can be recognized by iron response proteins and is commonly associated with iron-dependent oxidative stress. Iron restriction causes anemia, decreases liver iron levels, reduces MDA formation, and subsequently alters the expression of these IRPs.

Iron deposition in multiple tissues, including the brain, with aging is thought to contribute to the development of age-related diseases (*Zecca et al., 2004*; *Ward et al., 2014*). Iron dyshomeostasis in

the brain is an early feature of cognitive decline, such as in PD, and contributes to neuroinflammation, proteostasis failure, and other physiological abnormalities. There remains an unmet need to deeply understand the influence of reduced ETC activity on iron homeostasis in normative aging and neurodegeneration. Our data suggest that decreased Complex I activity with aging may help contribute to this neurodegeneration with brain iron accumulation, and therapeutic interventions targeting genetic or age-progressive causes of Complex I deficiencies may be especially beneficial.

The mitochondrial disease Leigh Syndrome is caused by mutations in more than 75 nuclear- or mitochondrial-encoded genes such as several subunits of Complex I (e.g., NDUFS4), Complex IV (e.g., COX10), assembly factors (e.g., SURF1), metabolic enzymes (PDHA1), and transport proteins (SLC19A3) among others (*Stenton and Prokisch, 2020*). It is critical to better understand the generality of pathogenic iron dyshomeostasis in Leigh Syndrome patients with these specific mutations. This work may have broader implications in the development of diagnostic markers for Leigh Syndrome, in which serum iron, transferrin saturation, serum ferritin, or serum hepcidin can be readily analyzed clinically. It may also lead to further beneficial treatment regimens in Leigh Syndrome or other mitochondrial diseases.

# Materials and methods

**Key resources table**

| Reagent type (species) or resource | Designation | Source or reference | Identifiers | Additional information |
|---|---|---|---|---|
| Strain, strain background (*Mus musculus*) | *Ndufs4<sup>tm1.1Rpa</sup>* C57Bl/6NCrl | Palmiter Laboratory *Kruse et al., 2008* | | |
| Other | PicoLab Mouse Diet 20 | LabDiet | Cat. 5058 | Facility Chow, Mouse Diet Studies |
| Other | AIN-93G Growth Purified Diet | LabDiet | Cat. 57W5 | Normal Iron (40 ppm), Mouse Diet Studies |
| Other | AIN-93G Growth Purified Diet | LabDiet | Cat. 5SSU | Low Iron (8 ppm), Mouse Diet Studies |
| Chemical compound, drug | Deferiprone (3-hydroxy-1,2-dimethyl-4(1H)-pyridone) | Sigma-Aldrich | Cat. 379409 | CAS 30652-11-0 |
| Chemical compound, drug | Ferric hydroxide dextran complex | Sigma-Aldrich | Cat. D8517 | CAS 9004-66-4 |
| Chemical compound, drug | Trace metal grade concentrated $HNO_3$ | Thermo Fisher Scientific | Cat. A509P500 | CAS 7697-37-2 |
| Chemical compound, drug | Low trace metals 30% $H_2O_2$ solution | Thermo Fisher Scientific | Cat. NC1199178 | |
| Chemical compound, drug | Ultra Trace Elemental Analysis Grade $H_2O$ | Thermo Fisher Scientific | Cat. W9-500 | |
| Chemical compound, drug | Ferrozine iron reagent, hydrate, 98% pure | Thermo Fisher Scientific | Cat. AC410570010 | CAS 1266615-85-3 |
| Chemical compound, drug | Trichloroacetic acid, 99% | Thermo Fisher Scientific | Cat. AAA1115636 | CAS 76-03-9 |
| Chemical compound, drug | Thioglycolic acid | Thermo Fisher Scientific | Cat. AAB2039122 | CAS 68-11-1 |
| Chemical compound, drug | HALT Protease and Phosphatase Inhibitor Cocktail (100X) | Thermo Fisher Scientific | Cat. 78444 | |
| Other | Bovine Serum Albumin, Heat Shock Treated | Thermo Fisher Scientific | BP1600-100 | Western Blot Assays |
| Chemical compound, drug | RIPA Lysis Buffer | Thermo Fisher Scientific | Cat. 89901 | |

*Continued on next page*

*Continued*

| Reagent type (species) or resource | Designation | Source or reference | Identifiers | Additional information |
|---|---|---|---|---|
| Chemical compound, drug | RestorePlus Stripping Buffer | Thermo Fisher Scientific | Cat. 46430 | |
| Commercial assay or kit | TBARS Assay Kit | Cayman Chemical | Cat. 10009055 | |
| Commercial assay or kit | SuperSignal West Pico PLUS Chemiluminescent Substrate | Thermo Fisher Scientific | Cat. 34578 | |
| Commercial assay or kit | SuperSignal West Femto Maximum Sensitivity Substrate | Thermo Fisher Scientific | Cat. 34095 | |
| Commercial assay or kit | Pierce BCA Protein Assay Kit | Thermo Fisher Scientific | Cat. 23225 | |
| Commercial assay or kit | PureLink RNA Mini Kit | Thermo Fisher Scientific | Cat. 12183025 | |
| Commercial assay or kit | iTaq Universal SYBR Green One-Step Kit | Bio-Rad | Cat. 1725151 | |
| Commercial assay or kit | Phire Tissue Direct PCR Master Mix | Thermo Fisher Scientific | Cat. F170L | Genotyping |
| Commercial assay or kit | No-Stain Protein Labeling Reagent | Thermo Fisher Scientific | Cat. A44717 | |
| Antibody | Anti-FTH1 (rabbit polyclonal) | Cell Signaling Technology | Cat. cs-3998 | 1:5000 |
| Antibody | Anti-TFR1 (rabbit monoclonal) | Abcam | Cat. ab214039 | 1:3000 |
| Antibody | Anti-FPN1 (rabbit polyclonal) | Thermo Fisher Scientific | Cat. PA5-77470 | 1:3000 |
| Antibody | Anti-DMT1 (mouse monoclonal) | Santa Cruz Biotechnology | Cat. sc-166884 | 1:3000 |
| Antibody | Anti-IRP1 (rabbit monoclonal) | Cell Signaling Technology | Cat. cs-20272 | 1:1000 |
| Antibody | Anti-IRP2 (rabbit monoclonal) | Cell Signaling Technology | Cat. cs-37135 | 1:1000 in 1% BSA |
| Antibody | Anti-GFAP (rabbit monoclonal) | Cell Signaling Technology | Cat. cs-12389 | 1:5000 |
| Antibody | Anti-Actin HRP conjugate (rabbit monoclonal) | Cell Signaling Technology | Cat. cs-5125 | 1:5000 |
| Antibody | Anti-rabbit IgG (H+L) secondary antibody, HRP (donkey polyclonal) | Thermo Fisher Scientific | Cat. 31458 | 1:20,000 |
| Other | m-IgGκ binding protein HRP conjugate | Santa Cruz Biotechnology | Cat. sc-516102 | 1:2000, Western Blot Assays |
| Sequence-based reagent, primer | Forward: TCAAGCCAGATCAGCATTCTC Reverse: AGCCAGTTTCATCTCCACATG | Integrated DNA Technologies | | *Tfr1* |
| Sequence-based reagent, primer | Forward: TCCTCATCACCATCGCAGACACTT Reverse: TCCAAACGTGAGGGCCATGATAGT | Integrated DNA Technologies | | *Dmt1A/B+IRE* *Dmt1A/B-IRE* |
| Sequence-based reagent, primer | Forward: TGGATGGGTCCTTACTGTCTGCTAC Reverse: TGCTAATCTGCTCCTGTTTTCTCC | Integrated DNA Technologies | | *Fpn1* |
| Sequence-based reagent, primer | Forward: CTCATGAGGAGAGGGAGCAT Reverse: GTGCACACTCCATTGCATTC | Integrated DNA Technologies | | *Fth1* |
| Sequence-based reagent, primer | Forward: GTCCCGTGGATCTGTGTCT Reverse: AGGAGCTAACCGCGAAGAGA | Integrated DNA Technologies | | *Ftl1* |

*Continued on next page*

*Continued*

| Reagent type (species) or resource | Designation | Source or reference | Identifiers | Additional information |
|---|---|---|---|---|
| Sequence-based reagent, primer | Forward: AAGCAGGGCAGACATTGCGAT<br>Reverse: CAGGATGTGGCTCTAGGCTATGT | Integrated DNA Technologies | | *Hamp* |
| Sequence-based reagent, primer | Forward: GTGTGAACGGATTTGGCCGTATTGGGCG<br>Reverse: TCGCTCCTGGAAGATGGTGATGGGC | Integrated DNA Technologies | | *Gapdh* |
| Sequence-based reagent, primer | Forward: GCTGAGAGGGAAATCGTGCGTG<br>Reverse: CCAGGGAGGAAGAGGATGCGG | Integrated DNA Technologies | | *Actb* |
| Other | Blood collection tube | Thermo Fisher Scientific | Cat. 02-669-33 | Lavendar Cap, CBC Assay |
| Other | Metal-Free Centrifuge Tube, 15 mL | VWR | Cat. 89049-172 | ICP-MS Assay |

## Animals and animal use

Experiments, procedures, and protocols described herein to care for, and handle, mice were reviewed and approved (protocol 4359-03) by the University of Washington Institutional Animal Care and Use Committee (IACUC) and strictly adhered to guidelines described in the Guide for the Care and Use of Laboratory Animals of the National Institutes of Health. Mice were housed at the University of Washington on a 14:10 hr light/dark cycle and provided food, water, and/or gel ad libitum. *Ndufs4*$^{+/-}$ were donated by the Palmiter Laboratory at the University of Washington and serially backcrossed with C57Bl/6NCrl mice. Pups produced by pairwise breeding of *Ndufs4*$^{+/-}$ were genotyped by PCR before post-natal day 21 (PND21) to identify *Ndufs4*$^{+/+}$, *Ndufs4*$^{+/-}$, and *Ndufs4*$^{-/-}$ pups. Male and female mice were distributed evenly in experiments, and *Ndufs4*$^{-/-}$ mice were housed with at least one healthy littermate for companionship and warmth. No more than five mice were housed per cage. Mice were monitored daily for common signs of neurodegeneration (e.g., clasping), weight loss, and moribundity. Mice that reached endpoint criteria, including >30% decrease from maximum body weight, loss of righting reflex, or were unresponsive to stimuli, were euthanized via primary and secondary $CO_2$ asphyxiation.

Mice were weaned around P21 and fed standard facility chow (PicoLab Mouse Diet 20 5058) or synthetic AIN-93G Growth Purified Diet containing normal (40 ppm, Lab Diet 57W5) or low (8 ppm, LabDiet 5SSU) amounts of iron. Mice fed the chow diet were treated with deferiprone in the water (0.5, 1, or 2 mg/mL water). Food and water were replenished every ~5 days. Mice on the control synthetic AIN-93G diet were injected every 3 days via intraperitoneal (i.p.) injection with iron-dextran (100 mg/kg body weight in saline) that was made from a stock containing 100 mg/mL iron. The volume injected was 6.67 μL/g body weight and mice were injected on alternate sides to minimize irritation or distress.

## Statistical analysis

The data presented in this study represent the weighted mean ± SEM with a minimum of three biological replicates and multiple technical replicates whenever possible unless otherwise indicated. Data or specimens were randomized and analyzed blindly and independently by multiple participants when possible. All studies utilized both male and female mice and sample size was chosen according to previous precedent (*Grillo et al., 2021*). Outliers were excluded if appropriately determined by Grubb's Test, interquartile range, or other methods when relevant. Statistical analysis for lifespan data was performed using a log-rank test, and p values for other experiments were calculated using student t test with Bonferroni correction or using ANOVA with post hoc Tukey test to account for multiple comparisons where appropriate. Pearson's test was used to test the significance of correlations. – $p<0.1$, *$p<0.05$, **$p<0.01$, ***$p<0.001$, ****$p<0.0001$, NS = Not Significant.

## Tissue collection

Tissues were freshly collected for protein, mRNA, and metabolic experiments from non-fasted ~35-day-old mice in the morning. Mice were euthanized via cervical dislocation and tissues were excised and flash frozen in liquid $N_2$ before processing. Tissues for analysis of metal content were obtained

and flash frozen from non-fasted ~35-day-old mice after anesthetization with a ketamine/xylazine cocktail and blood exhaustively removed via whole body perfusion with PBS.

## Complete blood count

Blood was collected via cardiac puncture from non-fasted ~30-day-old mice after anesthetization with a ketamine/xylazine cocktail. Blood was immediately transferred to a lavender cap blood collection tube containing EDTA to prevent coagulation. The blood was stored on ice and immediately submitted to the Veterinary Diagnostics Laboratory at the University of Washington for CBC analysis.

## Analysis of total metal content via ICP-MS

Tissues were collected after extensive perfusion as described above in non-fasted PND35 mice and flash-frozen in liquid $N_2$. Tissues were thoroughly homogenized in liquid $N_2$ and aliquoted into pre-tared metal-free Eppendorf tubes. Tissues were dried via a SpeedVac vacuum concentrator and weighed. Tissues were then digested with $HNO_3/H_2O_2$ after slowly heating to 95°C overnight. Digestates were then diluted in trace metal-free water containing dilute $HNO_3$ and submitted to the trace element analysis laboratory at the University of Washington for ICP-MS quantification. Any glassware or non-trace metal grade containers were washed with HCl acid solution to remove trace metals.

## Quantification of non-Hheme iron with ferrozine

A modified ferrozine-based assay (*Rebouche et al., 2004*) was used to quantify levels of non-heme iron in PND35 livers. Either standards (prepared from $FeSO_4$) or homogenized liver tissues were digested with 3 M HCl containing 600 mM trichloroacetic acid (TCA). Samples were heated at 65°C for 20 hr, centrifuged, and supernatant collected. Chromogen solution (100 µL containing 500 µM ferrozine, 1.5 M sodium acetate, 1.5% v/v thioglycolic acid in trace metal-free $H_2O$) was added to digestate (50 µL) in a 96-well plate and incubated for 30 min at room temperature to form a purple-colored solution. Ferrozine was quantified via measuring absorbance at 595 nM with a microplate reader and non-heme iron was calculated using the obtained standard curve.

## Quantification of MDA-TBA adduct

Lipid peroxidation was assessed by a malondialdehyde thiobarbituric acid (MDA-TBA) reactive substance assay using a modified procedure from the Cayman Chemicals MDA-TBA Assay Kit. Homogenized livers from PND35 mice were lysed using RIPA buffer containing protease inhibitors and supernatant collected after centrifugation. MDA-TBA adduct was prepared according to the manufacturer's instructions, except we used half the recommended volumes to increase signal. We read the absorbance of the MDA-TBA adduct at 530 nM and calculated MDA-TBA concentrations using a standard curve.

## Protein analysis via western blot

Proteins were extracted from homogenized liver and brains from PND35 mice with RIPA buffer containing protease and phosphatase inhibitors. Protein extract was collected after centrifugation, quantified by a BCA assay, and diluted to either 1 or 2 mg/mL. Samples for western blot were prepared by diluting protein (20 µg) in Laemmli sample buffer, reducing agent, and RIPA buffer. Samples were denatured at 95°C for 5 min and loaded onto NuPAGE 4–12% MIDI or MINI gels and ran at 120 V in MOPS running buffer. Protein was transferred to a PVDF membrane using a Trans-Blot Turbo Transfer system with manufacturer transfer buffer containing an additional 0.2% SDS. Total protein was imaged according to manufacturer protocol (Thermo Fisher Scientific A44717), and blots were blocked with either 5% BSA or 5% milk in TBST at room temperature for 1 hr. Membranes were incubated in either 1% or 5% BSA with primary antibodies overnight at 4°C, followed by 2 hr at room temperature. Membranes were rinsed with TBST, and incubated with secondary antibody for 1–2 hr in 5% BSA or 5% milk. After rinsing membranes with TBST, either Thermo Fisher SuperSignal PicoPlus or Femto ECL reagent was added and chemiluminescence detected with an iBright CL1500 system. Membranes were stripped for 10 min, rinsed, blocked, and re-probed as described above. Densitometry was performed using the iBright Analysis Software relative to actin loading control or total protein, and normalized to WT levels.

### mRNA quantification via qRT-PCR

Total RNA was isolated from PND35 livers using the Thermo Fisher PureLink RNA Mini Kit according to the manufacturer's instructions and quantified with a NanoDrop spectrophotometer. Primers reported from the literature were checked by NCBI Primer Blast for accuracy. Primers were purchased from Integrative DNA Technologies as RxnReady Oligos and diluted to a 1 mM stock. Expression of mRNA was quantified by qRT-PCR using the BioRad iTaq Universal SYBR Green One-Step RT-PCR Kit containing 300 nM primer and 50 ng/μL of total RNA in a 10 μL reaction. PCR reaction products were verified by melting temperature or gel electrophoresis and compared to expected band size. mRNA expression relative to *Gapdh* or *Actb* was normalized to WT levels.

## Acknowledgements

The authors thank Sarah Proffitt and Kerrie Allen for their assistance with CBC analysis through the University of Washington Veterinary Diagnostics Laboratory. The authors thank Alex Gagnon and Tamas Ugrai from the University of Washington Trace Element Laboratory for Environmental Science for quantification of metal content by ICP-MS. The authors gratefully acknowledge Silvan Urfer, Will Rieger, Tom Milstein, and Sanchita Narayan for assistance with animal husbandry and genotyping. ASG was supported by NIH NINDS Grant F32 NS110109. This work was supported by NIH NINDS Grant R01 NS098329 and NIH NIA Grant P30 AG013280 to MK.

## Additional information

### Funding

| Funder | Grant reference number | Author |
| --- | --- | --- |
| National Institutes of Health | F32 NS110109 | Anthony S Grillo |
| National Institutes of Health | R01 NS098329 | Matt Kaeberlein |
| National Institutes of Health | P30 AG013280 | Matt Kaeberlein |

The funders had no role in study design, data collection and interpretation, or the decision to submit the work for publication.

### Author contributions

CJ Kelly, Formal analysis, Validation, Investigation, Visualization, Methodology, Writing – original draft, Writing – review and editing; Reid K Couch, Formal analysis, Validation, Investigation, Writing – review and editing; Vivian T Ha, Formal analysis, Validation, Investigation, Visualization, Methodology, Writing – review and editing; Camille M Bodart, Validation, Investigation, Methodology, Writing – review and editing; Judy Wu, Sydney Huff, Investigation, Writing – review and editing; Nicole T Herrel, Validation, Investigation, Writing – review and editing; Hyunsung D Kim, Azaad O Zimmermann, Jessica Shattuck, Yu-Chen Pan, Investigation; Matt Kaeberlein, Conceptualization, Resources, Formal analysis, Supervision, Funding acquisition, Visualization, Writing – original draft, Project administration, Writing – review and editing; Anthony S Grillo, Conceptualization, Formal analysis, Supervision, Funding acquisition, Validation, Investigation, Visualization, Methodology, Writing – original draft, Project administration, Writing – review and editing

### Author ORCIDs

Matt Kaeberlein ⓘD http://orcid.org/0000-0002-1311-3421
Anthony S Grillo ⓘD http://orcid.org/0000-0003-3283-6585

### Ethics

Experiments, procedures, and protocols described herein to care for, and handle, mice were reviewed and approved (protocol 4359-03) by the University of Washington Institutional Animal Care and Use

Committee (IACUC) and strictly adhered to guidelines described in the Guide for the Care and Use of Laboratory Animals of the National Institutes of Health.

### Decision letter and Author response

Decision letter https://doi.org/10.7554/eLife.75825.sa1
Author response https://doi.org/10.7554/eLife.75825.sa2

## Additional files

### Supplementary files
• MDAR checklist

### Data availability

All data generated or analyzed during this study are included in the manuscript and supporting files or are publicly available via Dryad (https://doi.org/10.5061/dryad.xpnvx0khb).

The following dataset was generated:

| Author(s) | Year | Dataset title | Dataset URL | Database and Identifier |
|---|---|---|---|---|
| Grillo AS | 2023 | Data from: Iron Status Influences Mitochondrial Disease Progression in Complex I-Deficient Mice | https://doi.org/ 10.5061/dryad. xpnvx0khb | Dryad Digital Repository, 10.5061/dryad.xpnvx0khb |

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
