## [Editor Report]

This is an important study showing that the perturbation of NDUFS4, a component of mitochondrial respiratory complex I, which is also a key consumer of iron in the cell, can perturb iron metabolism, causing hepatic iron overload and neurological dysfunction. Compellingly, iron chelation reverses some of these pathological phenotypes. This paper will be of broad interest particularly to the neurology and iron biology communities, for its novel observations and experimental rigor.

---

## [Decision Letter]

**Decision letter after peer review:**

Thank you for submitting your article "Iron Status Influences Mitochondrial Disease Progression in Complex I-Deficient Mice" for consideration by *eLife*. Your article has been reviewed by 3 peer reviewers, and the evaluation has been overseen by a Reviewing Editor and Carlos Isales as the Senior Editor. The following individual involved in review of your submission has agreed to reveal their identity: Martin Picard (Reviewer #2).

The reviewers have discussed their reviews with one another, and the Reviewing Editor has drafted this to help you prepare a revised submission. In particular, input from Reviewer 1 and 3 should be addressed either by textual edits or experiments in the resubmission. Since there was no change in iron levels, FTH1 or TFR1 expression in the brain, it leaves open the possibility that a non-cell autonomous mechanism may be responsible for the neurological phenotypes. Addressing this by determining if a blood brain barrier-impermeable iron chelator can rescue these phenotypes will go a long way to addressing this question.

Essential revisions:

(1) The authors suggest that clasping is a good readout of neurodegeneration, but it would be nice to show this histologically or through a second, orthogonal method. The authors previously used histology to probe neuroinflammation Ndufs4 knockout brain tissue using GFAP staining (Martin-Perez, Grillo, and Ito, et al., Nat Metab, 2019). The authors could perform these experiments in wild type and Ndufs4 KO mice on both normal Fe and low Fe diets (or +/- chelator). The outcome of these experiments is likely to be informative regarding the mechanism, regardless of the result.

(2) The authors note that the chelating agent that they use, deferiprone (DFP), is "brain-penetrating." This raises the possibility that iron chelation in the brain may be critical for the observed phenotypes, despite the lack of evidence for altered iron responses at the molecular level. Some studies have suggested that other commonly used iron chelation agents, such as DFO (desferroxamine) or other hexadenate iron chelators, have poor solubility across the blood brain barrier (BBB) due to their hydrophilicity (reviewed in Liu G. et al., Biochim Biophys Acta, 2005 PMID 16051470). The authors could repeat the iron chelation experiments (readout could be clasping or lifespan extension, whatever is easiest) with an iron chelator that cannot efficiently cross the BBB. Should this chelation strategy still rescue the observed neuronal phenotypes/lifespan extension, this would support the non-local model for rescue. If this experiment fails to rescue the phenotypes in the knockouts mice, it suggests that local iron chelation in the brain is critical for the neurodegeneration and/or lifespan rescues, despite not seeing large effects in iron accumulation and signaling in the brain tissue of Ndufs4 KO mice.

---

## [Author Response]

Essential revisions:(1) The authors suggest that clasping is a good readout of neurodegeneration, but it would be nice to show this histologically or through a second, orthogonal method. The authors previously used histology to probe neuroinflammation Ndufs4 knockout brain tissue using GFAP staining (Martin-Perez, Grillo, and Ito, et al., Nat Metab, 2019). The authors could perform these experiments in wild type and Ndufs4 KO mice on both normal Fe and low Fe diets (or +/- chelator). The outcome of these experiments is likely to be informative regarding the mechanism, regardless of the result.

We agree with the reviewers that the suggested experiments would be informative regardless of the result. As suggested, we performed identical immunoblot experiments to previously reported (PMID 33324011) to evaluate changes in GFAP expression. NDUFS4-KO mice normally exhibit brain lesions in the olfactory bulb and cerebellum while most other brain regions are unaffected. We thus isolated olfactory bulb, cerebellum, and cortex from WT and NDUFS4-KO mice fed normal Fe or low Fe diets. Consistent with previous work, we observed significant increases in GFAP expression in the olfactory bulb and cerebellum in KO mice fed the normal Fe diet relative to control WT mice. We further observed that iron restriction reduces GFAP expression in these regions, consistent with the delay in clasping in mice fed a low Fe diet. Collectively, this data strengthens our conclusions that iron perturbations influence the neuroinflammatory phenotype in NDUFS4-KO mice. We made changes to the text and figures to reflect these additional experiments. =

(2) The authors note that the chelating agent that they use, deferiprone (DFP), is "brain-penetrating." This raises the possibility that iron chelation in the brain may be critical for the observed phenotypes, despite the lack of evidence for altered iron responses at the molecular level. Some studies have suggested that other commonly used iron chelation agents, such as DFO (desferroxamine) or other hexadenate iron chelators, have poor solubility across the blood brain barrier (BBB) due to their hydrophilicity (reviewed in Liu G. et al., Biochim Biophys Acta, 2005 PMID 16051470). The authors could repeat the iron chelation experiments (readout could be clasping or lifespan extension, whatever is easiest) with an iron chelator that cannot efficiently cross the BBB. Should this chelation strategy still rescue the observed neuronal phenotypes/lifespan extension, this would support the non-local model for rescue. If this experiment fails to rescue the phenotypes in the knockouts mice, it suggests that local iron chelation in the brain is critical for the neurodegeneration and/or lifespan rescues, despite not seeing large effects in iron accumulation and signaling in the brain tissue of Ndufs4 KO mice.

We thank the reviewers for the excellent suggestion to test the effects of a brain impermeable iron chelator (deferoxamine) on lifespan or the clasping phenotype. This experiment would provide strong support for or against brain non-autonomous effects in this mitochondrial disease model. We thus evaluated both clasping and lifespan in NDUFS4-KO mice treated with deferoxamine at or above doses previously reported (e.g PMID 19234060) to reduce murine body iron status (i.e. daily i.p. injection at 125 mg/kg).We found that treatment with this brain impermeable iron chelator was not able to delay disease progression. This data suggests that brain iron restriction is critical for the observed effects reported in the manuscript. =